

# A signature of tumor DNA repair genes associated with the prognosis of surgically-resected lung adenocarcinoma

Xiongtao Yang[1], Guohui Wang[2], Runchuan Gu[3], Xiaohong Xu[1] and Guangying Zhu[4]

[1] Department of Radiation Oncology, Peking University China-Japan Friendship School of Clinical Medicine, Beijing, China
[2] Department of Radiotherapy, Second Hospital of Hebei Medical University, Shijiazhuang, Hebei, China
[3] Department of Radiation Oncology, China-Japan Friendship Institute of Clinical Medicine, Beijing, China
[4] Department of Radiation Oncology, China-Japan Friendship Hospital, Beijing, China

Corresponding author
Guangying Zhu, zryyfa@163.com

## ABSTRACT

**Background**. Lung cancer has the highest morbidity and mortality of cancers worldwide. Lung adenocarcinoma (LUAD) is the most common pathological subtype of lung cancer and surgery is its most common treatment. The dysregulated expression of DNA repair genes is found in a variety of cancers and has been shown to affect the origin and progression of these diseases. However, the function of DNA repair genes in surgically-treated LUAD is unclear.

**Methods**. We sought to determine the association between the signature of DNA repair genes for patients with surgical LUAD and their overall prognosis. We obtained gene expression data and corresponding clinical information of LUAD from The Cancer Genome Atlas (TCGA) database. The differently expressed DNA repair genes of surgically-treated LUAD and normal tissues were identified using the Wilcoxon rank-sum test. We used uni- and multivariate Cox regression analyses to shrink the aberrantly expressed genes, which were then used to construct the prognostic signature and the risk score formula associated with the independent prognosis of surgically-treated LUAD. We used Kaplan–Meier and Cox hazard ratio analyses to confirm the diagnostic and prognostic roles. Two validation sets (GSE31210 and GSE37745) were downloaded from the Gene Expression Omnibus (GEO) and were used to externally verify the prognostic value of the signature. OSluca online database verifies the hazard ratio for the DNA repair genes by which the signature was constructed. We investigated the correlation between the signature of the DNA repair genes and the clinical parameters. The potential molecular mechanisms and pathways of the prognostic signature were explored using Gene Set Enrichment Analysis (GSEA).

**Results**. We determined the prognostic signature based on six DNA repair genes (PLK1, FOXM1, PTTG1, CCNO, HIST3H2A, and BLM) and calculated the risk score based on this formula. Patients with surgically-treated LUAD were divided into high-risk and low-risk groups according to the median risk score. The high-risk group showed poorer overall survival than the low-risk group; the signature was used as an independent prognostic indicator and had a greater prognostic value in surgically-treated LUAD. The prognostic value was replicated in GSE31210 and GSE37745. OSluca online database analysis shows that six DNA repair genes were associated with poor prognosis in most lung cancer datasets. The prognostic signature risk score correlated with the

pathological stage and smoking status in surgically-treated LUAD. The GSEA of the risk signature in high-risk patients showed pathways associated with the cell cycle, oocyte meiosis, mismatch repair, homologous recombination, and nucleotide excision repair. **Conclusions**. A six-DNA repair gene signature was determined using TCGA data mining and GEO data verification. The gene signature may serve as a novel prognostic biomarker and therapeutic target for surgically-treated LUAD.

# INTRODUCTION

Lung cancer has the highest morbidity and mortality of cancers worldwide (*Siegel, Miller & Jemal, 2020*). Lung adenocarcinoma (LUAD) is the most common pathological subtype of lung cancer, accounting for approximately 40% of all cases (*Denisenko, Budkevich & Zhivotovsky, 2018*). Surgery is the most common treatment for LUAD (*Hirsch et al., 2017*) although there is a high rate of relapse with 30–40% of patients dying from stage I surgically-treated LUAD (*Oskarsdottir et al., 2016*; *Westaway et al., 2013*). The 5-year overall survival (OS) rate decreases to 36% for those with stage III–IV surgically-treated LUAD (*Mansuet-Lupo et al., 2014*). The patients with surgically resected LUAD had varied prognoses, but patients with partially surgically-resected cancers had poor prognoses, regardless of whether the patients were early-LUAD treated with surgery alone or advanced-LUAD undergoing aggressive postoperative therapy (including platinum-based chemotherapy and radiation therapy). The biological mechanism of these differences is not fully understood (*De Miguel-Perez et al., 2019*). It is important to determine the prognoses of surgically-treated LUAD patients to provide specific treatments for improved OS.

Previous studies have shown that DNA repair systems play an important role in cancer (*Gavande et al., 2016*; *Jeggo, Pearl & Carr, 2016*; *Mei et al., 2019*; *Tessoulin et al., 2018*). Five major DNA repair pathways are vital to maintaining the genetic stability of cells (*Chatterjee & Walker, 2017*): base excision repair (BER), nucleotide excision repair (NER), mismatch repair (MMR), homologous recombination (HR) and non-homologous end joining (NHEJ), two specific lesion repair pathways (direct chemical reversal and interstrand crosslink (ICL) repair) are important supplements for DNA damage repair. DNA repair genes play an essential role in tumorigenesis, tumor progression, and response to therapy(*Lima et al., 2019*). Efforts are being made to identify DNA damage repair genes as therapeutic targets and to develop agents against these targets. For example, poly ADP-ribose polymerase 1 (PARP1) is involved in multiple DNA repair pathways to maintain genomic stability so PARP1 inhibitors have been approved to treat ovarian cancer, breast cancer, and other DNA repair-deficient tumors (*Ray Chaudhuri & Nussenzweig, 2017*). LUAD exists in multiple DNA repair gene mutations but no studies have been conducted on DNA damage repair pathways in LUAD to date. We studied surgically-resected LUAD
and the prognostic and predictive significance of the expression of DNA repair genes for this cancer.

We constructed a multi-gene signature suitable for clinical application in surgical LUAD patients, based on multi-gene signatures that indicate a poor prognosis in high-risk populations of specific tumor patients. This approach increases our knowledge of tumor cell generation, growth, and metastasis, and creates new avenues for targeted therapy. We obtained 201 surgically-treated LUAD cases from The Cancer Genome Atlas (TCGA) database and extracted and identified key mRNAs associated with DNA repair and established a six-gene risk signature that could accurately predict the prognosis of these specific patients.

## MATERIALS & METHODS

### Data source and preprocessing

We extracted the gene expression profiles and corresponding clinical information of patients with LUAD from the TCGA database (https://portal.gdc.cancer.gov/). We obtained the gene expression profiles of 59 normal samples and screened 535 LUAD samples for performance status, OS, survival state, and gene expression profiles. We collected data from 201 surgically-resected LUAD cases. The gene mRNA expression data were normalized using the Transcripts Per Kilobase of exon model per Million mapped reads (TPM) method (*Li et al., 2010*). Clinical information including gender, age, pathological stage and TNM stage were included in this research. The Gene Expression Omnibus (GEO) database (https://www.ncbi.nlm.nih.gov/geo/) for microarray-based expression data was used to obtain data from patients with surgically-resected LUAD, and that datasets to be considered must be containing more than 100 cases with information of OS and living status. The expression data of the GSE31210 set were normalized by the MAS5 algorithm and transformed with log2. The expression data of the GSE37745 set were normalized by the Robust Multi-Array Average (RMA) method and transformed with log2. 226 cases from GSE31210 (validation set 1) and 106 cases from GSE37745 (validation set 2) were screened with gene expression data and relevant clinical information. The OSluca online database (http://bioinfo.henu.edu.cn/LUCA/LUCAList.jsp) contains multiple lung cancer datasets (*Yan et al., 2020*), which can perform hazard ratio analysis on screened prognostic genes related to DNA repair. TCGA data were applied as a training set. GEO data sets were applied as validation sets. OS was defined as the date of the first treatment to the date of death of any cause or to the time of the last follow-up. The clinical characteristics are summarized in Table 1.

### Bioinformatic analysis

Differential expression genes (DEGs) were filtered according to the adjusted $p$-value < 0.05, |log2fold change| > 1.5 in the surgically-treated LUAD and normal samples of the training set. DNA repair genes were further screened from DEGs. Univariate Cox regression analysis evaluated the DNA repair genes significantly related to OS according to $p$-value < 0.05. Multivariate Cox regression analysis was used for further screening. DNA repair genes are executors in the extremely complicated processes of DNA damage repairing which remedies

**Table 1  Clinic pathological characteristics of surgical LUAD patients from the training (TCGA) and-validation sets (GSE31210, GSE37745).**

| Characteristics | TCGA (n = 260) | | GSE31210 (n = 226) | | GSE37745 (n = 106) | |
|---|---|---|---|---|---|---|
| | Number of cases % | | Number of cases % | | Number of cases % | |
| Platform | | | | GPL570 | | GPL570 |
| Noncancerous samples | | 59 | | 0 | | 0 |
| Surgically-resected LUAD samples | | 201 | | 226 | | 106 |
| Age (years) | | | | | | |
| ≥65 | 99 | 49.3 | 62 | 27.4 | 52 | 49.1 |
| < 65 | 97 | 48.3 | 164 | 72.6 | 54 | 50.9 |
| Gender | | | | | | |
| Male | 91 | 45.3 | 105 | 46.5 | 46 | 43.4 |
| Female | 110 | 54.7 | 121 | 53.5 | 60 | 56.6 |
| Stage | | | | | | |
| Stage 2-4 | 95 | 47.3 | 58 | 25.7 | 36 | 34.0 |
| Stage 1 | 106 | 52.7 | 168 | 74.3 | 70 | 66.0 |
| T | | | | | | |
| T3-4 | 30 | 14.9 | | | | |
| T1-2 | 169 | 84.1 | | | | |
| N | | | | | | |
| N1-3 | 66 | 32.8 | | | | |
| N0 | 128 | 63.7 | | | | |
| M | | | | | | |
| M1-Mx | 57 | 28.4 | | | | |
| M0 | 144 | 71.6 | | | | |
| Smoking | | | | | | |
| Yes | | | 111 | 49.1 | | |
| No | | | 115 | 50.9 | | |

**Notes.**

Abbreviations: LUAD, lung adenocarcinoma; T, tumor size; N, lymph node status; M, metastasis.

specific types of DNA damage through specific pathways. We obtained a prognostic signature of six DNA repair genes and a corresponding prognostic risk score formula based on a linear combination of corresponding multivariate Cox regression coefficients. The formula for the risk score was: expression level of gene $1 \times K1$ + expression level of gene $2 \times K2 + \cdots +$ expression of level gene n $\times Kn$. The risk score for every surgically-treated LUAD patient was calculated using this formula and the population was categorized into high-risk and low-risk groups according to the median risk score value. We used gene set enrichment analysis (GSEA) (https://www.gsea-msigdb.org/gsea/msigdb) to identify pathways associated with the high-risk and the low-risk groups of DNA repair gene signatures. The Kyoto Encyclopedia of Genes and Genomes (KEGG) gene sets were downloaded from the MSigDB database. Enrichment false discovery rates (FDR) were based on 1,000 permutations. The high-risk score vs. low-risk score and FDR < 0.07 were considered to be statistically significant.

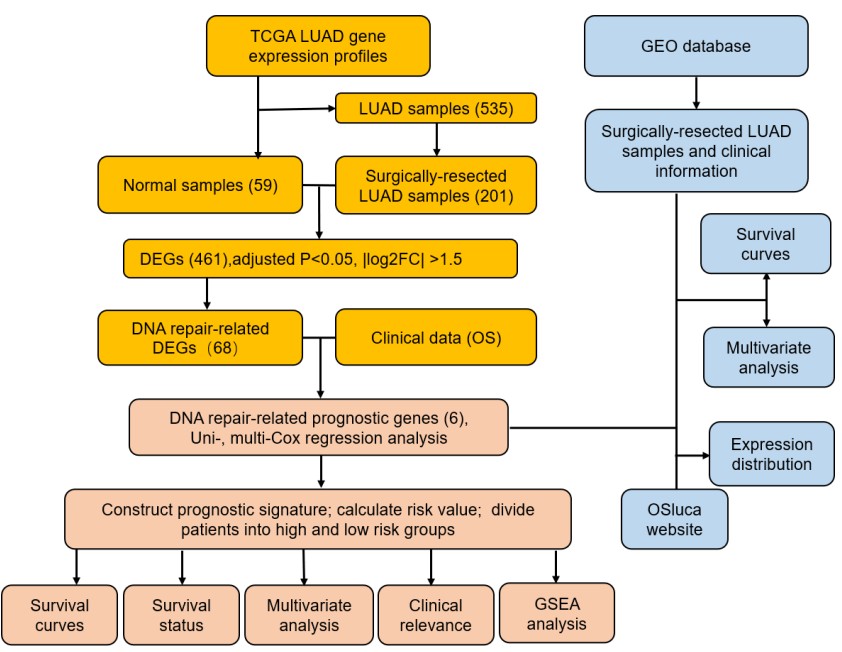

**Figure 1  Flow diagram of methods for developing and verifying the six DNA repair-related prognostic genes signature.**

## Statistical analysis

The gene expression differences between the surgically-treated LUAD group and the normal group were compared using the unpaired two-tailed student's $t$-test. We evaluated the prognostic value of the risk score and clinicopathological features in patients with surgically-treated LUAD using univariate and multivariate Cox analysis. The Kaplan–Meier method was used to analyze the correlation between the risk score and OS. The survival curves were compared using a log-rank test. All statistical analyses were performed using R 3.6.3 (https://www.r-project.org/) and SPSS 20.0.

## RESULTS

### Identification of DNA repair-related prognostic genes

Study was conducted in accordance of the flow chart (Fig. 1). LUAD gene expression data and clinical records were downloaded from TCGA database. We screened DEGs from surgically-treated LUAD and normal samples of the training dataset. In accordance with common senses and definition, we collected DNA repair-related gene sets from the MSigDB database and literature records (Table S1) (*Pearl et al., 2015*), and further selected from the DEGs to identify the DNA repair-related prognostic gene signature for surgically-treated LUAD. We acquired 68 genes related to DNA repair (Fig. 2A). Among them, the expression levels of four genes in surgical LUAD were lower than normal samples, while the other genes were the opposite. (Figs. 2B, 2C). These genes were used in Univariate Cox regression analysis and resulted in 23 genes significantly correlated with OS (Fig. 2D). The 23 genes were screened by multivariate Cox regression analysis, resulting in a total of six genes,

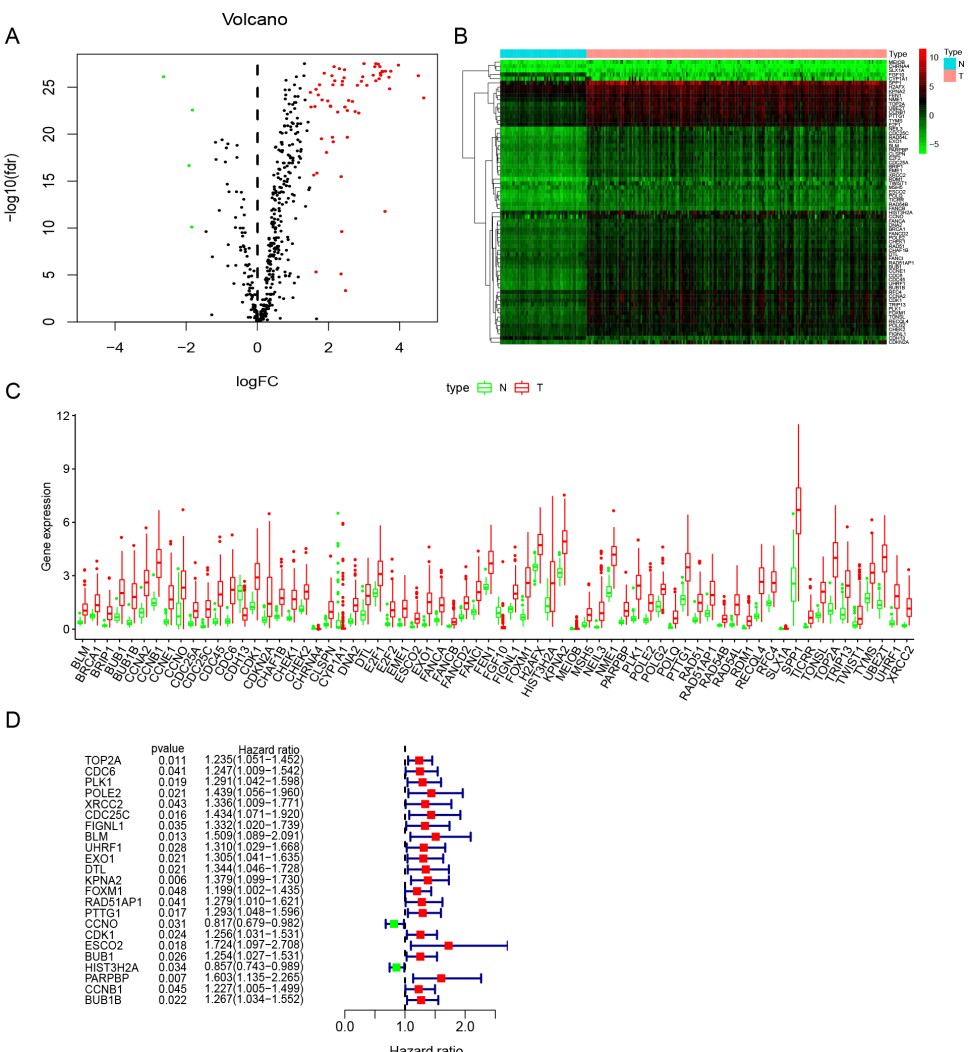

**Figure 2** **Identification of DNA repair-related prognostic genes.** (A) Volcano plot of DNA repair-related DEGs. | log2(Fold Change)| > 1.5 and FDR < 0.05 were set as screening criteria. The green, red and black dots represented the down-, up-regulated DNA repair-related DEGs and genes that were not satisfied the screening criteria, respectively. (B) The heatmap of all 68 DNA repair-related DEGs distributions. (C) The Boxplot of all 68 DNA repair-related DEGs distributions. (D) The forest plot of 23 prognostic DNA repair-related DEGs screened out by univariate Cox regression. DEGs, Differentially expressed genes; N, Normal samples; T, Surgical lung adenocarcinoma.

namely PLK1, FOXM1, PTTG1, CCNO, HIST3H2A and BLM. A prognostic signature was constructed (Table 2) based on the six DNA repair-related prognostic genes.

## The performance of DNA repair-related prognostic gene signature

The risk score for surgically-treated LUAD patients was calculated based on the gene mRNA level and corresponding coefficients using the formula: Risk score = (0.555713) x PLK1 expression value + (−0.5696) × FOXM1 expression value + (0.291471) × PTTG1 expression value + (−0.30485) × CCNO expression value + (−0.17162) × HIST3H2A

**Table 2** The information of 6 prognostic mRNAs importantly associated with overall survival inpatients with surgical LUAD.

| mRNA | Ensemble ID | Location | Coefficient | HR | *P*-value |
|---|---|---|---|---|---|
| PLK1 | ENSG00000166851 | Chr16:23,677,656-23,690,367 | 0.555713 | 1.743183 | 0.043687 |
| FOXM1 | ENSG00000111206 | Chr12: 2,857,681-2,877,155 | −0.5696 | 0.565754 | 0.024525 |
| PTTG1 | ENSG00000164611 | Chr5: 160,421,855-160,428,739 | 0.291471 | 1.338395 | 0.064214 |
| CCNO | ENSG00000152669 | Chr5: 55,231,152-55,233,608 | −0.30485 | 0.737234 | 0.004893 |
| HIST3H2A | ENSG00000181218 | Chr1: 228,456,979-228,457,873 | −0.17162 | 0.842298 | 0.030497 |
| BLM | ENSG00000197299 | Chr15: 90,717,346-90,816,166 | 0.442472 | 1.556551 | 0.119493 |

Notes.
Abbreviations: LUAD, lung adenocarcinoma; HR, hazard ratio; ID, identity.

expression value + (0.442472) × BLM expression value. The patients were divided into high-risk and low-risk groups based on the median risk score. The patients' risk score distribution and OS are shown in Figs. 3A and 3B. The expression profile distribution of six DNA repair-related genes is shown in Fig. 3C. A Kaplan–Meier curve showed poor prognoses for the high-risk group (log-rank test *p*-value < 0.0001; Fig. 3D). We used univariate and multivariate Cox hazard ratio analysis to further assess the performance of our signature risk score in surgically-treated LUAD patients based on other common clinicopathological features. Univariate Cox regression analysis showed that tumor stage and N classification were significantly correlated with survival. N classification was significantly correlated with survival in multivariate Cox regression analysis. Remarkably, univariate and multivariate Cox analysis showed that the risk score was more significantly correlated with survival, giving it a greater prognostic value (Figs. 3E and 3F). These results indicated that the six-gene signature risk score could reliably predict the survival of surgical LUAD patients.

## Validation of the six-gene signature for survival prediction

Gene expression data and clinical information of LUAD patients with surgical records and survival times were selected from the GEO database to further prove the accuracy of our gene signature. We obtained the GSE31210 and GSE37745 data sets. The GSE31210 expression data were standardization and distributed as shown in Figs. 4A and 4B. The GSE37745 set contained data from pathological samples of squamous cell lung carcinoma, lung adenocarcinoma, and large-cell carcinoma. We screened surgically-treated LUAD samples; their standardization and distribution is shown in Figs. 4C and 4D. The DNA repair-related signature and risk score formulas were applied to verify the two validation sets and the Kaplan–Meier curve showed that the high-risk group had poor OS when compared with the low-risk group (Fig. 4E *p*-value = 0.008, Fig. 4F *p*-value = 0.003). Apart from that, multivariate Cox hazard ratio regression analysis in the validation sets GSE31210 and GSE37745 had also been conducted. According to the results, risk score can be considered as an independent risk factor (Fig. 4G HR = 1.523, *p*-value = 0.023) in the GSE37745 set, but since only clinical stage I and II patients are included in the GSE31210 set differing from the training set TCGA which contains clinical I, II, III, IV stages, it could be insufficient regarding risk score as an independent risk factor after the inclusion of stage staging (Fig. 4H HR = 1.083, *p*-value = 0.072). The six DNA repair-related genes of the

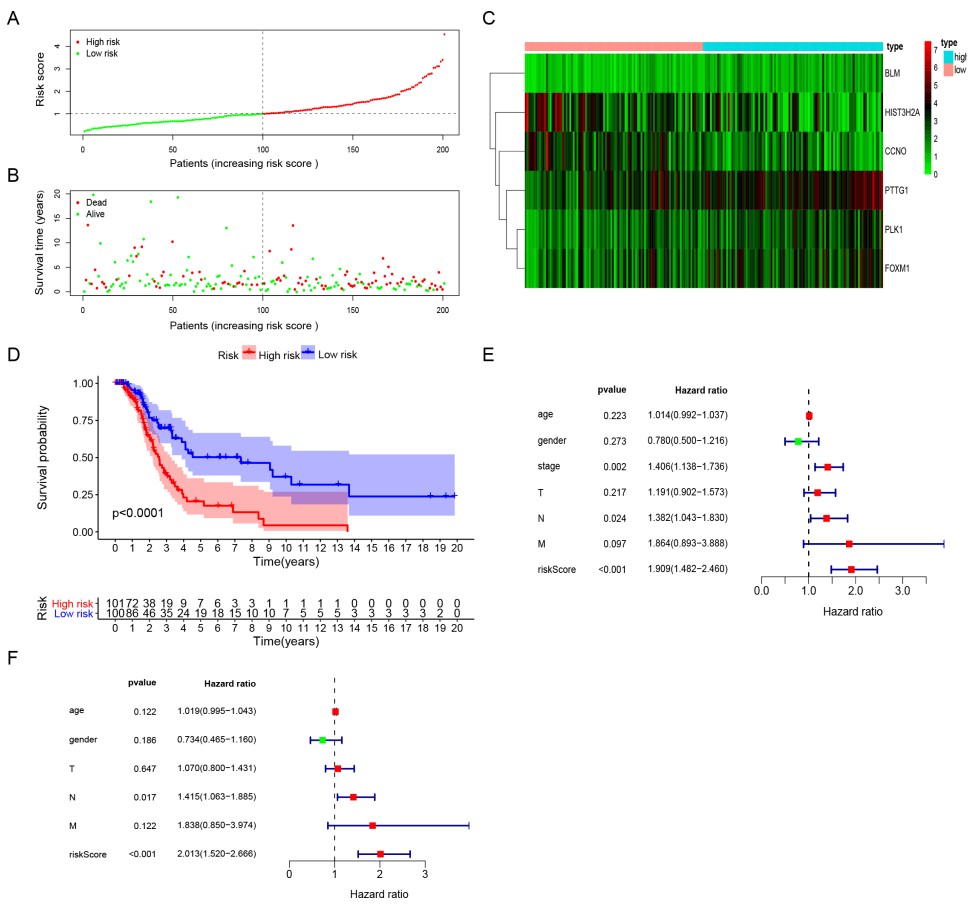

**Figure 3** **The performance of DNA repair-related prognostic gene signature.** (A) Six prognostic DNA repair-related DEGs screened out by multivariate Cox regression to construct signature and calculate risk score, the distribution of risk score of each patient. (B) The distribution of survival status of each patient. (C) The heatmap of Six DNA repair-related signature DEGs distributions in risk score groups. (D) Kaplan–Meier survival curves for overall survival in the DNA repair-related risk score groups. (E) Univariate Cox hazard ratio analysis shown the signature risk score and other clinicopathological features related to overall survival of surgical LUAD. (F) Multivariate Cox hazard ratio analysis shown the signature risk score and other clinicopathological features related to overall survival of surgical LUAD. High: high risk; Low: low risk; gender:0 (female); 1 (male); stage: 1, 2, 3, 4; T:1, 2, 3, 4;N:0, 1, 2, 3, M:0, 1.

signature were analyzed in the OSluca online database for forest plot prognostic analysis. The results showed that these 6 genes were associated with poor prognosis in most lung cancer data sets (Fig. 5, HR > 1). The above results indicated that the signature had greater external validity and reliability.

## Relationship between six-gene signature risk score and clinical parameters

We studied the correlation between the six-gene signature risk score and the clinical parameters under the training and validation sets. The sets were divided into two subgroups that included age, gender, pathological stage, TNM stage, and smoking status. The stratification results are shown in Table 1. In the training set, parameters such as age,

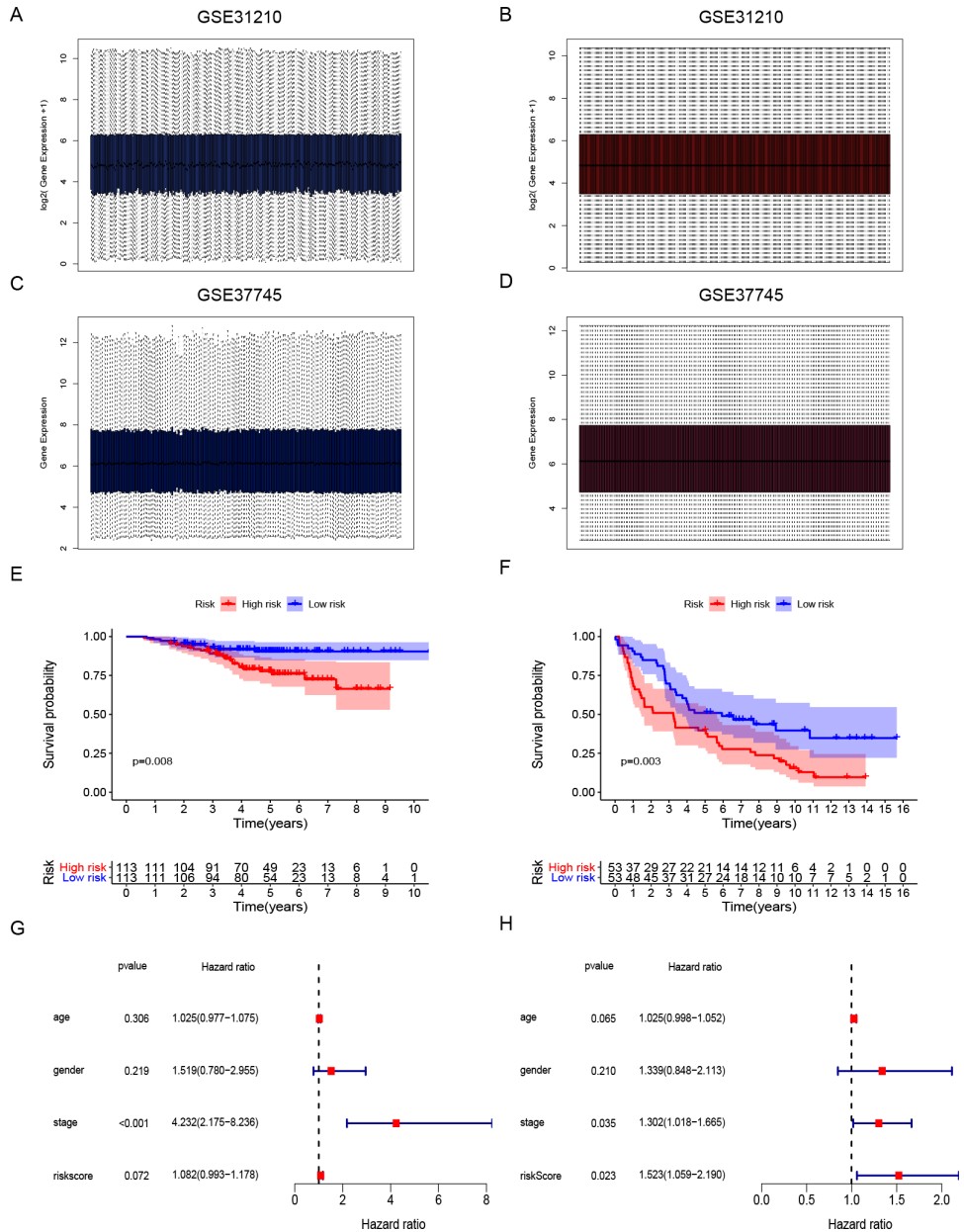

**Figure 4** **Development and validation of DNA repair-related signature for survival prediction.** (A–B) The GSE31210 expression data were standardization and distribution. (C–D) The GSE37745 expression data were standardization and distribution. (E) Kaplan-Meier survival curves of overall survival in the DNA repair-related risk score groups in GSE31210 set. (F) Kaplan-Meier survival curves of overall survival in the DNA repair-related risk score groups in GSE37745 set. (G) Multivariate Cox hazard ratio analysis shown the signature risk score and other clinicopathological features related to overall survival of GSE31210. (H) Multivariate Cox hazard ratio analysis shown the signature risk score and other clinicopathological features related to overall survival of GSE37745. gender:0 (female); 1 (male); stage: 1, 2, 3, 4.

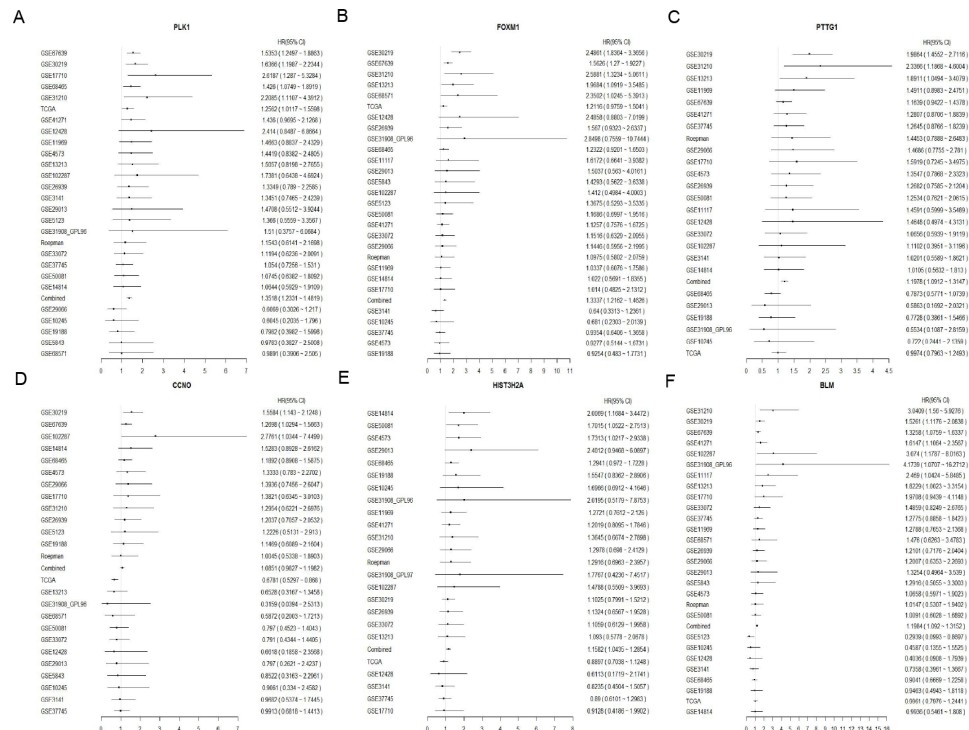

**Figure 5 Forest plot of six DNA repair-related gene in Osluca online database.** The hazard ratio of each DNA repair-related gene of the signature in the multiple lung cancer datasets (A) PLK1, (B) FOXM1, (C) PTTG1, (D) CCNO, (E) HIST3H2A, (F) BLM.

gender, TNM stage did not affect risk score except for the correlation between pathological stage and risk score (Fig. 6A) (*p*-value = 0.022). In the GSE31210 set, the pathological stage and smoking status were correlated with the risk score, but age and gender were not related to the risk score (Fig. 6B) (*p*-value = 0.0004, *p*-value = 0.027, respectively). In the GSE37745 set, age, gender and pathological stage were not related to the risk score (Fig. 6C).

## The biological processes and pathways associated with the six-gene signature

The population was divided into a high-risk group and a low-risk group based on the risk score of DNA repair gene expression and the prognosis between the two groups was significantly different. We examined the whole gene expression profile in TCGA data using GSEA analysis to reveal the potential mechanism of the six-gene signature, which indicated that the main enrichment pathways were the cell cycle, oocyte meiosis, DNA replication, mismatch repair, homologous recombination and nucleotide excision repair pathways. Our results suggested that the poor prognosis of patients with surgically-resected LUAD was related to the DNA damage repair pathway (Fig. 7).

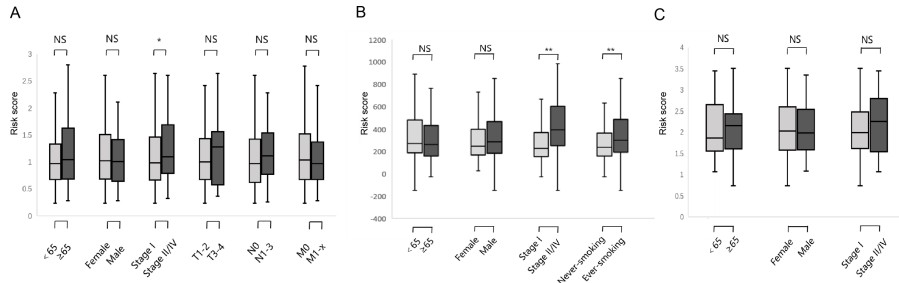

**Figure 6  Relationship between six DNA repair-related gene signature risk score and clinicopathological features.** (A) The relationship between risk score distribution and clinical parameters stratification. Including age, gender, tumor pathologic stage, T stage, N stage and M stage in TCGA surgical LUAD. (B) The relationship between risk score distribution and clinical parameters stratification. Including age, gender, tumor pathologic stage and smoking in GSE31210 surgical LUAD. (C) The relationship between risk score distribution and clinical parameters stratification. Including age, gender, and tumor pathologic in GSE37745 surgical LUAD. *Represent for *p*-value < 0.05; *Represent for *p*-value < 0.01; NS represent no statistical difference.

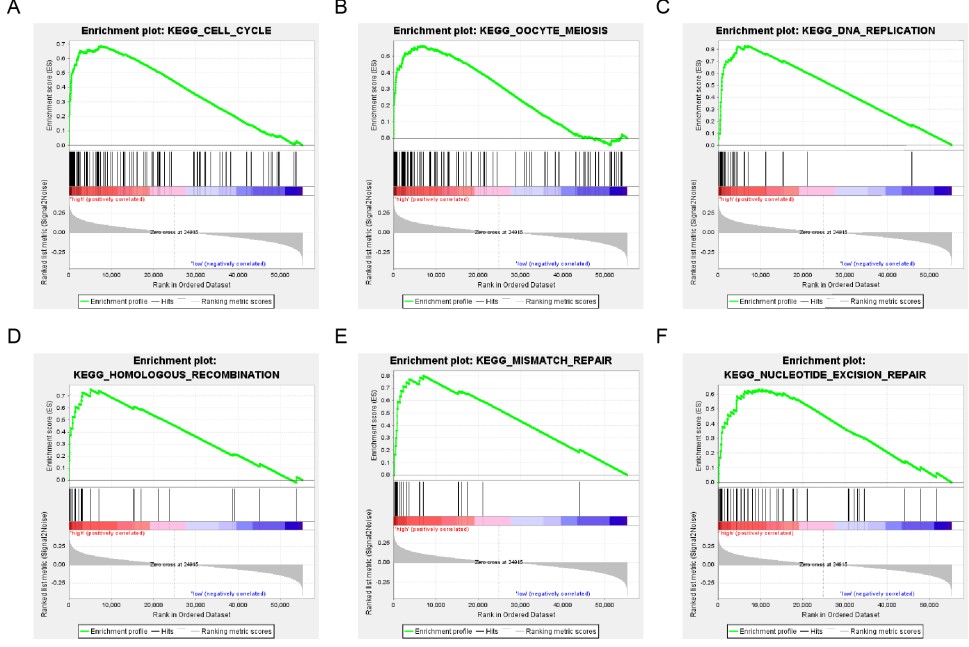

**Figure 7  six gene sets significantly enriched in the surgical LUAD phenotype using GSEA.** Including (A) cell cycle, (B) oocyte meiosis, (C) DNA replication, (D) homologous recombination, (E) mismatch repair and (F) nucleotide excision repair pathway.

## DISCUSSION

Lung cancer has the highest mortality rate of malignant tumors and adenocarcinoma is the most common histological subtype (*Network, 2014*). Video-assisted thoracoscopic surgical lobectomy for early-stage NSCLC and robotic lobectomy for advanced NSCLC is

the primary surgical treatment for LUAD (*Cerfolio et al., 2018*; *Chen et al., 2013*; *Veronesi et al., 2018*). Advancements in imaging techniques, diagnostic methods, surgical skills, and anesthesia management have led to an increase in the number of LUAD patients who were evaluated for and underwent surgery (*Ucvet, Gursoy & Yazgan, 2020*). Surgical treatment of patients with LUAD can extend survival time but there may be limited benefits to the partial surgical treatment of LUAD patients because of local recurrence and distant metastasis. Therefore, effective prognostic and therapeutic strategies to improve the survival rate of surgical LUAD are being actively explored. The development of high-throughput sequencing technology has improved the identification of potential biomarkers for tumor progression and prognosis. For example, Xu et.al. (2020) confirmed that flamingo subfamily Cadherin EGF LAG seven-pass G-type receptor 2 (CELSR2) is significantly overexpressed in hepatocellular carcinoma and may serve as a novel prognostic biomarker. Feng et al. (2020) used the Cox proportional hazards model to confirm that MHC class I chain-related B (MICB) expression was significantly associated with clinical parameters in colorectal cancer (CRC) and high MICB expression was an independent protective factor for OS. However, the predictive power of such biomarkers is limited in context-specific cancers because it is influenced by many factors, including transcription modification, translation modification, and temporal and spatial heterogeneity. Thus, a signature based on multi-gene cooperation can improve the predictive ability of each gene, resulting in a more reliable and robust predictive ability versus a single biomarker (*Ashley, 2015*; *Jiang et al., 2020*; *Wolfe et al., 2020*; *Zhu et al., 2016*). DNA damage repair genes are needed to maintain cell homeostasis and are extensively involved in cell replication, chromatin modification, DNA repair, checkpoint signaling, reactive oxygen species, metabolism, senescence, and apoptosis cellular mechanisms (*Brinkman, Liu & Kron, 2020*; *Csiszar et al., 2019*; *Gonzalo & Coll-Bonfill, 2019*; *Lee et al., 2018*; *Royce, Brown-Borg & Deepa, 2019*; *Ungvari et al., 2019*). Genetic mutations can effectively induce tumorigenesis and progression (*Pearl et al., 2015*). Targeted inhibition of the function of specific genes in DNA damage repair response has potential for clinical applications and the development of targeted drugs (*Li et al., 2019*; *Nickoloff et al., 2017*; *O'Connor, 2015*; *Tarantini et al., 2019*). We used DNA repair genes to construct and validate a prognostic signature that successfully predicts the survival time of surgically-treated LUAD patients.

We filtered the data downloaded from the TCGA database to obtain gene expression profiles and clinical data of patients with surgically-treated LUAD. The standardized DNA repair gene DEGs were screened for OS-related univariate and multivariate Cox regression analysis and the DNA repair-related prognosis signature and risk score calculation formulas were obtained. We divided the surgically-treated LUAD patients into high- and low-risk groups according to the calculated formula. Kaplan–Meier survival analysis showed that the high-risk group had a poor prognosis and Cox analysis showed that the risk score could be used as an independent prognostic factor, verifying the predictive effect of the signature. The GSE31210 and GSE377745 sets downloaded from GEO were used to perform survival analysis and the results were consistent with those from the Kaplan–Meier and Cox analyses. The DNA repair-related gene signature was shown to be highly reliable. GSEA

analysis suggested that the poor prognosis of patients with surgically-resected LUAD was related to the DNA damage repair pathway, according to *p*-value < 0.05 and FDR < 0.07.

We obtained six DNA repair genes making up the prognostic signature. Some of the six genes (PLK1, FOXM1, PTTG1, CCNO, HIST3H2A and BLM) were associated with tumorigenesis. PLK1 is a member of the Polo-like kinases (PLKs) family of serine/threonine protein kinases and plays an essential role in many stages of mitosis. The expression of PLK1 is cell cycle-dependent and its expression peaks in the M phase of a normal cell. PLK1 is highly expressed in many human cancers and its overexpression is associated with a poor prognosis. Previous studies have shown that the PLK1 protein interacts with multiple tumor suppressors to cause tumorigenesis and that targeting the protein can improve the sensitivity of tumors to chemoradiotherapy. The target PLK1 inhibitor has shown considerable promise in clinical studies and is being tested in clinical trials (*Gutteridge et al., 2016*; *Liu, Sun & Wang, 2017*). FOXM1 is a proliferation specific transcriptional modulator and is involved in the cell cycle, mitotic spindle integrity, angiogenesis, metastasis, apoptosis, and DNA damage repair pathway. This protein dysfunction contributes to tumorigenesis (*Nandi et al., 2018*). A meta-analysis reported that elevated FOXM1-protein expression was significantly associated with poor survival in most solid tumors. This suggests that FOXM1 is a potential biomarker for predicting prognoses in solid human tumors (*Li et al., 2017*). PTTG1 has been identified as a critical signature gene associated with tumor metastasis. It is overexpressed in a variety of endocrine-related tumors especially those of the pituitary, breast, and testis, as well as in nonendocrine-related cancers involving the pulmonary and gastrointestinal systems (*Vlotides, Eigler & Melmed, 2007*). Studies have shown that PTTG1 is highly expressed in breast cancer and PTTG1 may increase breast cancer cell growth through the nuclear exclusion of p27 (*Xiea & Wangb, 2016*). BLM is a member of the RecQ helicases, and can unwind forked dsDNA and anneal ssDNA, and play critical roles in DNA repair, recombination, replication, and transcription. The mutations of BLM cause Bloom syndrome (*Croteau et al., 2014*). The small molecule inhibitor of BLM, ML216, exhibits cell-based activity and can induce sister chromatid exchanges, enhance the toxicity of aphidicolin, and exert antiproliferative activity in cells expressing BLM (*Nguyen et al., 2013*). CCNO and HIST3H2A were less prevalent in the high-risk group and there are few reports about the two genes. CCNO is part of the cyclin family and is an essential regulator of endogenous apoptosis and participates in DNA damage repair (*Krokan & Bjoras, 2013*). HIST3H2A is a replication-dependent histone H2A gene located in the HIST3 cluster on chromosome 1 in the human genome sequence. We integrated the six DNA repair genes into a signature and found that patients in the high-risk group were associated with worse survival, which confirms the predictive value of the DNA repair gene signature in surgically-treated LUAD. The development of drugs against these target genes may be used as a potential treatment in surgically-treated LUAD.

We filtered out patients with combined radiotherapy, chemotherapy, neoadjuvant therapy and postoperative adjuvant therapy within the pre-set framework. In TCGA, due to the small number of LUAD cases with surgical treatment combined with radiotherapy, neoadjuvant treatment or adjuvant treatment, we did not conduct further analysis. 69 cases were found with chemotherapy and the survival rate of low-risk group was higher,

but no significant difference was obtained ($p = 0.17$) due to the small size of the sample, follow-up needs to expand the case for verification. Tumor markers, as a class of substances synthesized and released by tumor cells, or increased as the body reacting to tumor cells, are mainly applied for auxiliary diagnosis, prognostic judgment, curative effect observation and guiding follow-up treatment in current clinical practice. Although being convenient, highly-efficient, repeatably-detected and of low price, they are still limited by its relatively poor specificity and sensitivity. As specified for patients with LUAD after surgery, our multi-gene signature is supposed to have better specificity and the efficacy of radiotherapy can be predicted based on the expression of DNA repair-related genes for those with postoperative radiotherapy combined. Disadvantages mainly include the need for clinical sample sequencing which could be difficult to obtain and high sequencing cost.

We use data mining and bioinformatics analysis on the TCGA, GEO and OSluca databases to construct a DNA repair-related gene prognosis signature. Since the number of enrolled cases is relatively small, more cases should be recruited to expand the study. In addition, we need more clinical practice to further verify our conclusions.

## CONCLUSIONS

We used data from surgically-treated LUAD to identify DNA repair-related DEGs and constructed and validated a six-gene signature for predicting the outcomes of surgically-treated LUAD. Further study of these DNA repair-related genes showed that the prognosis-risk signature of DNA repair-related genes can be used as potential predictive markers and therapeutic targets for the surgical treatment of LUAD.

## ACKNOWLEDGEMENTS

Thanks to all the researchers and staff working for The Cancer Genome Atlas database and Gene Expression Omnibus Database.

### Funding

This study was supported by the grants from the National Key R&D Program of China (to Guangying Zhu) (No.2018YFC1313202) and the China-Japan Friendship Hospital Scientific Research Start-up Funds (to Guangying Zhu)(No.2016-RC-4). The funders had no role in study design, data collection and analysis, decision to publish, or preparation of the manuscript.

### Grant Disclosures

The following grant information was disclosed by the authors:
The National Key R&D Program of China: 2018YFC1313202.
The China-Japan Friendship Hospital Scientific Research Start-up Funds: 2016-RC-4.

### Competing Interests

The authors declare there are no competing interests.

## Author Contributions

- Xiongtao Yang conceived and designed the experiments, performed the experiments, analyzed the data, prepared figures and/or tables, authored or reviewed drafts of the paper, and approved the final draft.
- Guohui Wang conceived and designed the experiments, prepared figures and/or tables, and approved the final draft.
- Runchuan Gu performed the experiments, prepared figures and/or tables, and approved the final draft.
- Xiaohong Xu analyzed the data, prepared figures and/or tables, and approved the final draft.
- Guangying Zhu conceived and designed the experiments, authored or reviewed drafts of the paper, and approved the final draft.

## Data Availability

The raw data are available in the Supplementary Files.

## Supplemental Information

Supplemental information for this article can be found online at http://dx.doi.org/10.7717/peerj.10418#supplemental-information.

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
