# Peer review of "A signature of tumor DNA repair genes associated with the prognosis of surgically-resected lung adenocarcinoma"

_PeerJ, doi:10.7717/peerj.10418_

## Round 0.1 · original submission · Major Revisions

The manuscript you submitted to PeerJ has been reviewed. The reviewers have recommended publication pending major revisions. Therefore, I invite you to respond to the reviewers' comments at the bottom of this letter and revise your manuscript accordingly.

Reviewer 1 ·

Basic reporting

Lung cancer has the highest morbidity and mortality of cancer, so the authors use TCGA and GEO datasets dipping 6 novel signature of DNA repair genes for LUAD patients.

Experimental design

a. The sample and the profile cohorts in the study are small, suggest using more datasets to forecast those DNA repair genes as a prognostic and therapy target for LUAD using online tools, such as OSluca for lung cancer (http://bioinfo.henu.edu.cn/LUCA/LUCAList.jsp).
b. Test the relationship between DNA repair genes and therapy, such as radiation, molecular therapy or neoadjuvant therapy;
c. Suggest using OSluca tool to form a forest analysis of the signatures of LUAD;

Validity of the findings

Please using more datasets to assess the signatures of LUAD.

Additional comments

a. Please see the EXPERIMENTAL DESIGN.
b. Improve the manuscript.
c. discuss merit and demerit the signatures of DNA repair genes, compared to the clinically available biomarkers.

Reviewer 2 ·

Basic reporting

Yang et al. evaluated the prognostic impact of 6 DNA repair genes in resected lung cancer.

The aim is generally interesting. However, there are several issues to be solved or explained more for the publication.

Experimental design

First of all, authors chose 6 genes in considering the impact on OS. Therefore, it is out of question that these genes are related to prognosis. The GSE31210 and GSE37745 cohorts validated the results in TCGA data set using survival curves. However, multivariate analysis should be performed not only in TCGA cohort but in validation cohorts.

Figure 2
The location of C and D should be changed according to the alphabet order.
In multivariate analysis, the detail profile or cutoff point of each variable must be clarified. For example, T descriptors were divided into T1-2 and T3-4 as described in Table 1?
TNM stage is often affected by N or M descriptor (In M1-2 cases, the T and N descriptors do not matter for staging). So, fundamentally, these descriptors must not be handled with stage to avoid the double effect of N or M descriptor.

Validity of the findings

Figure 4
It is hard to understand the meaning to compare the prognosis between relapse and relapse-free cases. Authors estimated stage or smoking status using Kaplan-meire curves but the impact of 6 genes are obscure unless multivariate analysis is performed. The points should be explained clearly.

Additional comments

Please refer the comments above.

Reviewer 3 ·

Basic reporting

This manuscript reported DNA repair genes-based signature in surgically-resected lung adenocarcinoma. These findings are potential interesting and useful.

Experimental design

None

Validity of the findings

None

Additional comments

This manuscript reported DNA repair genes-based signature in surgically-resected lung adenocarcinoma. These findings are potential interesting and useful. However, a few points need to be better clarified.

1. The flowchart of the whole analysis process should be provided.

2. The authors should consider having a native English speaker, or English Language Editing Service – preferably with background in biology – to revise this work.

3. Please referring to the Table 1 published in paper [PMID: 30604627], a table summarizing the main characteristics of TCGA and GEO datasets should be provided.

4. The authors said “We acquired 68 genes related to DNA repair (Fig. 1).” The interpretation of Fig.1 is too simple, and needs more details.

5. The raw data of several figures needs to be uploaded as Supplementary Tables, such as Fig.1A, Fig.2D, Fig.3C, Fig.3D, etc.

6. The points as follows should be added the references. “DNA repair systems play an important role in cancer”(Line 64) [PMID: 30545397, 31018854]; “DNA repair genes play an essential role in tumorigenesis, tumor progression, and response to therapy”(Line 70) [PMID: 30975222]; “DNA damage repair genes are needed to maintain cell homeostasis…” (Line 222-224) [PMID: 31280482, 31037472, 31655958, 31721033, 29717417]; “Targeted inhibition of the function of specific genes in DNA damage repair…targeted drugs” (Line 226-227) [PMID: 31679124, 31521196].

7. Some limitations of this study should be provided in Discussion

8. What’s the inclusion criteria of screened potential GEO datasets?

Reviewer 4 ·

Basic reporting

Well written.

Experimental design

Well designed.

Validity of the findings

no comment.

Additional comments

The study identified a 6-DNA repair gene signature associated with prognosis of surgically-resected lung adenocarcinoma based on TCGA and GEO database, which may serve as a novel prognostic biomarker and therapeutic target.

The study is well designed and written. There are some considerations before a major revision.

1. Please define ‘DNA repair gene’ in ‘Materials & Methods’. Or it will confuse the readers.

2. Suggest the authors provide a research workflow.

3. Age was classified as ≥63 and <63. Stage was classified as Stage 1 and Stage 2-4. Why? Different classifications lead to different models.

4. The text in the tables and figures is too small. Note the numbering sequence of sub-tables.

5. Suggest to delete Figure 4(D-H), which seems to have nothing to do with the topic of the article

6. Suggest to delete ‘The mortality rate was 52.5% …the chi-square test (p-value=0.008)’. The chi-square test used here is incorrect. Survival analysis is better here for survival data.

7. The study mainly depends on bioinformatics analysis. More validation from external datasets can make it more reliable.

---

## Round 0.2 · Minor Revisions

Please revise the manuscript as the reviewer suggested.

Reviewer 2 ·

Basic reporting

The manuscript has been revised according to suggestions.

Experimental design

In revision, the defaults were revised or explained more persuasively.

Validity of the findings

no comment

Additional comments

The manuscript has been revised according to suggestions. The aim is generally interesting. Reviewer is looking forward to reading further related articles from same group in the future.

Reviewer 3 ·

Basic reporting

It is good.

Experimental design

none

Validity of the findings

none

Additional comments

none.

Reviewer 4 ·

Basic reporting

no comment

Experimental design

no comment

Validity of the findings

no comment

Additional comments

The authors responded well. However, it still needs some minor revision.

1. The authors selected 'DNA repair-related genes' according to Table S1. They had better describe the source of Table S1, and provide the reference source.

2. Suggest to keep the same font format in Tables and Figures. For instance, all fonts should be unified as Arial in Table 1 and Figure 1.

---

## Round 0.3 · accepted · Accept

After carefully revised, this manuscript has been greatly improved and now can be accepted.